# Food Insecurity: Is It a Threat to University Students’ Well-Being and Success?

**DOI:** 10.3390/ijerph18115627

**Published:** 2021-05-25

**Authors:** Nor Syaza Sofiah Ahmad, Norhasmah Sulaiman, Mohamad Fazli Sabri

**Affiliations:** 1Department of Nutrition, Faculty of Medicine and Health Sciences, Universiti Putra Malaysia, Serdang 43400, Selangor, Malaysia; fiqzatuanahmad@gmail.com; 2Research Centre of Excellence for Nutrition and Non-Communicable Diseases, Faculty of Medicine and Health Sciences, Universiti Putra Malaysia, Serdang 43400, Selangor, Malaysia; 3Malaysian Research Institute on Ageing (MyAgeing), Universiti Putra Malaysia, Serdang 43400, Selangor, Malaysia; 4Department of Resource Management and Consumer Studies, Faculty of Human Ecology, Universiti Putra Malaysia, Serdang 43400, Selangor, Malaysia; fazli@upm.edu.my

**Keywords:** food security, food insecurity, university students, academic performance, stress, anxiety, depression

## Abstract

Food insecurity is a growing concern among university students. The high prevalence of food insecurity is a threat to students’ health and success. Therefore, this study aims to determine an association between food security status, psychosocial factors, and academic performance among university students. A total of 663 undergraduate students in seven randomly selected faculties in Universiti Putra Malaysia participated in this study. An online survey was conducted to obtain demographic and socioeconomic characteristics, food security status (six-item USDA; food security survey module, FSSM), psychosocial factors (depression, anxiety and stress scale, DASS-21) and academic performance. Among the abovementioned participating students, 32.4% are male. About 62.8% reported to have experienced food insecurity. Binary logistic regression revealed that students whose fathers were working (AOR = 6.446, 95% CI: 1.22, 34.01) came from low- (AOR = 14.314, 95% CI: 1.565, 130.954) and middle-income groups (AOR = 15.687, 95% CI: 1.720, 143.092), and those receiving financial aid (AOR = 2.811, 95% CI: 1.602, 4.932) were associated with food insecurity. Additionally, food insecurity students were less-likely reported, with CGPA ≥ 3.7 (AOR = 0.363, 95% CI: 1.22–34.014). Food insecurity respondents had higher odds for stress (AOR = 1.562, 95% CI: 1.111, 2.192), anxiety (AOR = 3.046, 95% CI: 2.090, 4.441), and depression (AOR = 2.935, 95% CI: 2.074, 4.151). The higher institutions should identify students with food insecurity problems and future intervention programs need to be conducted to combat food insecurity among students, thus yielding benefits to their health and success.

## 1. Introduction

Presently, the increasing alarm on the high prevalence of food insecurity among university students has received much attention among researchers [1]. Food insecurity or a lack of “access to enough and nutritious food (at all times) for an active, healthy life” [2] has long been a global issue for vulnerable groups such as women, children, and the elderly. However, the recent concern on food insecurity among students has been identified as an emerging “skeleton in the university closet” [3]. Globally, a recent systematic review on college/university students in the United States revealed that 14% to 59% of them had undergone food insecurity, which exceeded the national prevalence (12.3%) [4]. Similarly, Malaysia’s experience was within the range of 22% to 69.4% of college/university students [5,6,7,8,9], which too exceeded the prevalence of food security at the national level (Peninsular Malaysia), which was 11.4% [10].

Previous empirical studies have documented several factors of food insecurity that affect students. Food insecurity is more likely to be experienced by students who come from a low-income family [1,8,9,10,11], male students [6,7] and those not living with their parents [11,12,13]. Surprisingly, students who receive financial aid are more likely to become food insecure [1,8,11]. The rise in tuition fees, insufficient financial aid, and high living costs have been suggested as possible reasons for students to experience food insecurity [1,5,8]. Financial problems faced by the students worsen their food insecurity conditions as they cannot afford to purchase enough, as well as nutritious food for their basic needs in order to become healthy and perform well in their studies [7].

In addition, a growing amount of literature suggested that food insecurity affects students negatively [14,15,16,17,18]. The students with food insecurity are reported as more likely to experience a poor academic performance. According to Morris et al. [14], students with a grade point average (GPA) less than 3.00 are more likely to experience food insecurity compared to those with a GPA above 3.00. Nutrition affects students’ thinking skills, behavior and health [15]. Inadequate macro- and micro-nutrients interferes with the students’ learning process, as they are unable to concentrate, hence affecting their academic performance [16]. 

Food insecurity is also associated with poor health. Food-insecure students were reported to have high odds of stress [17] and depression [17,18,19]. Financial difficulties and stressful life events have indeed affected university students’ well-being and academic performance [19]. Well-being is defined as the state of experiencing positive emotions and moods, the absence of negative emotions (depression and anxiety), feeling good and happy as well as having high life satisfaction [20]. Stress, anxiety and depression are types of mental health that include emotional and social well-being [21]. When one experiences stress, anxiety or depression, it becomes a threat to their well-being as they are unable to be happy and well [21]. 

Nevertheless, in Malaysia most studies have identified the factors associated with food insecurity, yet little is known about its consequences on students’ health and success. This study is an important step towards determining the negative impacts of food insecurity on students. Therefore, this study aims (1) to determine the factors associated with food security status among undergraduate students attending Universiti Putra Malaysia (UPM), Malaysia, and (2) to determine the association between food security status, psychosocial factors, and academic performance among undergraduate students attending Universiti Putra Malaysia (UPM), Malaysia.

## 2. Materials and Methods

### 2.1. Study Design

This study was a cross-sectional study conducted to investigate the relationship between food security status, psychosocial factors, and academic performance among undergraduate students of Universiti Putra Malaysia (UPM), Malaysia. UPM is a public research university in Malaysia located in Serdang, next to Malaysia’s administrative capital city, Putrajaya. Multi-stage samplings were used in this study, where seven courses in UPM were randomly selected, with participations being all second-year and third-year undergraduate students from selected courses. We selected seven courses to be included in our study based on an estimation calculation of students per course. First, we obtained the total enrollment of undergraduate students and courses in UPM via the UPM official website. From there, it was estimated that the total number of undergraduate students per course was 185 students and the estimated undergraduate students in the second and third year were 92 students. Next, the total number of courses was calculated by simply dividing the total sample size (645) with the estimated number of second and third year students (seven courses).

The seven faculties and courses involved were Faculty of Forestry (Bachelor of Forestry Science), Faculty of Medicine and Health Sciences (Bachelor of Science (Nutrition and Community Health)), Faculty of Science (Bachelor of Science in Biology with Honors), Faculty of Educational Studies (Bachelor Science (Home Science)), Faculty of Engineering (Bachelor of Engineering (Civil)), Faculty of Food Science and Technology (Bachelor of Science (Food Studies) with Honors) and Faculty of Biotechnology and Biomolecular Sciences (Bachelor of Science Biotechnology). As for participants, second-year and third-year students were chosen because they were deemed suitable to be participants as they had already spent one or two years to adapt with the university’s life. First year students were not included because they were still new with the transition from a dependent life before enrollment to an independent university life. In fact, they were still finding their way to adapt with the life in which they had to make decisions on their own. Meanwhile, fourth year students were not considered for this study because it was their final year where they were busy with their final year project and in fact, it was quite difficult to meet them. 

Data collection was carried out from May 2019 to January 2020 via online survey. Ethical approval was obtained from Ethics Committee for Research Involving Human Subjects, Universiti Putra Malaysia (UPM/TNCPI/RMC/1.4.18.2). An approval letter to conduct the research in selected faculties was obtained and the contact numbers of the students’ representatives from each course were acquired. Each representative was approached and briefed about the study, where he or she was then requested to share the link of the online questionnaire with all the classmates to be completed. In addition, the demographic and socioeconomic characteristics, food security status, psychosocial factors and academic performance of the respondents were acquired. Informed consent forms were also obtained from the respondents.

### 2.2. Measures

Demographic and socioeconomic characteristics data were collected included age, gender, ethnicity, level of education, marital status, parent’s educational background and occupation, monthly household income, financial aid recipient, total financial received and total expenses in one semester. 

US Department of Agriculture Food Security Survey Module (USDA-FSSM) was used to assess food security status among the respondents. The 6 items with the affirmative responses of “often”, “sometimes” and “yes” were scored as 1 meanwhile non-affirmatives responses of “never”, “no” and “don’t know” were scored as 0. The total score was 6 and respondents with a score of 2 or more indicated food insecurity. 

Psychosocial factors were assessed using depression, anxiety and stress scale (DASS-21). This scale consists of 21 items with three subscales as follows: depression (7 items), anxiety (7 items) and stress (7 items). Each item was scored on a scale of 0 (did not apply to me at all) until 3 (applied to me very much). The sum of every item was calculated and multiplied by 2. The possible total score was 42. The severity labels were classified as normal, moderate and severe. The classification could be helpful to characterize the degree of severity relative to population. For example, a moderate score indicated that the person was above the population mean, not the moderate level of the disorder. For moderate and severity levels, further clinical assessment was needed to determine appropriate diagnose and treatment.

Academic performance was self-reported by respondents to obtain the latest cumulative grade point average (CGPA). The CGPAs then were classified into first class honors (≥3.7) and honors (<3.7) according to UPM grading system.

### 2.3. Data Analysis

All data were analyzed by using IBM SPSS version 22 (IBM Corp, Armonk, NY, USA). All descriptive statistics were reported in mean and standards deviation for continuous data, meanwhile frequency and percentage for categorical data. For bivariate analysis, Chi-square was used to compare categorical variables. In addition, binary logistic regression (enter method) was used to investigate the relationship between food security status, psychosocial factors and academic performance. The variable with *p* < 0.25 in the Chi-square test were all included in the analysis. The results were presented in odds ratio (OR) with 95% confidence interval (CI). The level of statistical significance was set at *p* < 0.05 (i.e., the bivariate analysis and logistic regression). 

## 3. Results

A total of 663 undergraduate students participated in this study. The mean age of the respondents was 21.9 years, with the majority of them being female (67.6%) (Table 1). Almost all of the respondents lived independently (96.4%) and were not married (96.2%). Most of the respondents had a working father (82.5%) and over half of the respondents had a working mother (64.1%). In addition, the mean monthly household income was MYR 6746.65 (=USD 1619.20), with more than two-fifths of the respondents having an income below MYR 4850 (=USD 1164) (44.6%), and a smaller proportion of the respondents having a monthly household income more than MYR 10,959 (=USD 2630.16). Furthermore, 62.8% were food-insecure and for academic performance, only 11.4% had a CGPA more than or equal to 3.7 with the mean being 3.42. In the context of psychosocial factors, the mean for the stress, anxiety and depression scores were 25.50 ± 15.178, 25.82 ± 14.623 and 17.79 ± 15.494, respectively. A larger proportion of the respondents had normal stress (65.5%) and depression (59.9%), but not for anxiety (moderate level = 42.3%).

Table 2 shows the associations between demographic and socioeconomic, food security status, psychosocial factors and academic performance. There were no significant associations found between gender, living arrangement, and status with food security status (*p* > 0.05). Nevertheless, father (χ^2^ = 3.866, *p* < 0.05) and mother (χ^2^ = 5.086, *p* < 0.05) occupation, monthly household income (χ^2^ = 9.234, *p* < 0.05) and students who received financial aid (χ^2^ = 12.319, *p* < 0.001) were significantly associated with the food security status. Moreover, food security status was associated with academic performance (χ^2^ = 14.601, *p* < 0.001) and psychosocial factors, which were stress (χ^2^ = 8.733, *p* < 0.05), anxiety (χ^2^ = 38.936, *p* < 0.001) and depression (χ^2^ = 39.030, *p* < 0.001). 

### 3.1. Factors Associated with Food Security Status

The adjusted logistic regression revealed that the factors that had remained significantly associated were father occupation, monthly household income and financial aid receiver (*p* < 0.05) (Table 3). The students who had a working father were 6.5 times more likely to become food-insecure (AOR = 6.446, 95% CI: 1.22, 34.014). Those with a monthly household income below RM4850 (=USD 1164) (AOR = 14.314, 95% CI: 1.565, 130.954) and in between RM4850 (=USD 1164) to RM10959 (=USD 2630.16) (AOR = 15.687, 95% CI: 1.720, 143.092) were more prone to be food-insecure compared to those with a monthly household income above RM10959 (=USD 2630.16). Meanwhile, the students who received financial aid had higher odds to experience food insecurity (AOR = 2.811, 95% CI: 1.602, 4.932). 

### 3.2. Food Security Status Affect Psychosocial Status and Academic Performance

Additionally, food security status affect students’ psychosocial status and academic performance (Table 4). Food insecurity students were reported as less likely to get a pointer more than or equal to 3.7 for their academic performance (AOR = 0.363, 95% CI: 1.22–34.014) compared to food-secure students. On the other hand, food insecurity respondents were reported as having higher odds for stress (AOR = 1.562, 95% CI: 1.111, 2.192), anxiety (AOR = 3.046, 95% CI: 2.090, 4.441) and depression (AOR = 2.935, 95% CI: 2.074, 4.151).

## 4. Discussion

This work examined the factors associated with food security status among undergraduate students attending Universiti Putra Malaysia (UPM), Malaysia, as well as associations between food security status, psychosocial factors, and academic performance among them. Approximately 62.8% of the respondents were reported to have experienced food insecurity. The findings of this research were in line with the existing body of knowledge for the high prevalence of food insecurity among university students [5,6,7,8,9]. Furthermore, a recent study conducted among undergraduate students in four public universities across Peninsular Malaysia reported that 60.9% of students had experienced food insecurity [7]. According to Nurulhudha [7], the trend of high prevalence was more significant when involving multiple institutions rather than focusing on only one institution. Nevertheless, the aforementioned trend of the high prevalence of food insecurity among university students was also exhibited in this study even though it was conducted only in one institution. This indicates the researchers’ growing concern as more of the conducted studies among university students has revealed the hidden problem faced by them nowadays.

Moreover, this study revealed that a working father, the monthly household income and being a financial aid recipient are the factors associated with food security status. This study documented significant associations between students with a working father and food insecurity. There was a contradiction with the existing literature where no association was mostly found. The possible explanation was that the father’s income is not usually allocated for food, thus food insecurity prevails even if the father is working. Fathers normally spend their income on leisure, luxury or pleasure. On the contrary, a mother is more likely to allocate her money for health and food needed by the children compared to that of a father for a variety of cultural and biological reasons. For example, mothers who work have high self-esteem and control of the decision-making process for their children [22]. Previous studies have supported that children are less likely to experience food insecurity when the intra-household resources allocation is controlled by their mother [23]. However, one study by Ukegbu et al. [24] found a significant association between the type of the fathers’ occupations and the food security status. The students whose father works as a farmer are more likely to experience food insecurity compared to those whose father works as a salaryman. This is because the crops yielded by their father are sold to generate income instead of feeding their family [24].

In this study, students of low- and middle-income groups were associated with food insecurity. These findings were consistent with those of previous studies [9,24]. Low income is a major predictor of food insecurity across all vulnerable groups in relation to food insecurity. Students from low-income families spend less on food compared to those from high-income families [25]. In addition, healthy food costs more for students to purchase at all times, hence, they are more inclined to go for the more affordable ones [7]. In contrast, several studies conducted in Malaysia have reported that no association was found between income and food insecurity despite it being the major predictor for food insecurity [5,6,7]. On the other hand, financial aid recipients were also found to be associated with food security status in this study. Likewise, several foreign and local studies reported the same finding [1,8,11]. The high cost of living among the students, expensive nutritious food, rise in tuition fees, and inadequacy of financial aid received are the primary factors that can lead to such food insecurity [1,7,8,11]. 

According to a recent systematic review conducted in Malaysia regarding food insecurity, university students are suffering with their finances due to inadequate financial loans/scholarships and high living expenses [26]. Furthermore, poor financial management, too, can lead to not having enough money to buy food. The unplanned allocation of money and expenses can lead to the students’ losses; therefore, this will lead to unhealthy food choices. Besides that, in some cases, the students have extra money, but it is not spent for food. Instead, they spend it on gadgets or make up [27]. In order to help students with their financial burdens, higher institutions play their role by conducting seminars on financial management to provide the students with knowledge on how to use money wisely. On the other hand, UPM also gives their students the opportunity to have extra income by doing part-time job on campus, for example, working at the convenience stores within the campus or doing services such as printing and delivery. Besides that, UPM also provides a food bank to the students who are in need. The foods provided are usually the ones that last longer such as biscuits, instant food and 3-in-1 drinks. 

Furthermore, food insecurity negatively affects students’ health and academic performance. This study suggested that food insecurity was significantly associated with a low CGPA, as well as high stress, anxiety and depression levels. This study found that the food insecurity group had a low academic performance. This consistent finding with the previous literature indicates that food insecurity hinders a student’s success [14]. Morris et al. revealed that students with a GPA of more than 3.0 experienced higher food security compared to those with a low GPA range [14]. When the body does not obtain enough food, one becomes fatigued and experiences sleep deprivation and anxiety, where the body then becomes fragile, which interferes with the student’s ability to concentrate in class. The energy depletion from not having enough food can worsen the ability to perform well academically [13,14,28]. Furthermore, food insecurity disrupts the learning process because insufficient nutrient leads to low brain functioning [29]. 

Additionally, this study found that food insecurity interfered with the students’ wellbeing. Apparently, food-insecure students had higher odds for stress, anxiety and depression. This finding was in line with several past studies [18,19,30]. The inability of the students’ bodies to receive enough nutrition can cause emotional distress [31]. They become unhappy, thus their emotional and social well-being is disrupted, as well. Normally, food-insecure students become apprehensive because they do not have enough money to buy food. Thus, this situation can lead to stress as they struggle to feed themselves. As a result, eating unhealthy food and improper meals (because they are cheaper) may also interfere with their mental health. Hence, it is not surprising that food insecurity has been associated with high levels of stress, anxiety, and depression, and subsequently acts as a stressor that interrupts the physiological functioning and decreases metal health [32]. While well-being is defined as the experience of happiness, feeling well and the presence of good emotion, negative emotions such as stress, anxiety and depression disrupt the concept of an individual’s well-being [20].

The limitations of this study included the findings that could not represent the undergraduates of the whole country as this study only involved those of one particular university (i.e., Universiti Putra Malaysia). Besides that, the self-reported findings by the respondents produced biasness as they might not answer truthfully, especially on sensitive questions. Nevertheless, this limitation could be reduced by stating that all the data were confidential and used for research purposes only. Despite these limitations, this study contributes to the body of knowledge on food insecurity involving data of university students. Furthermore, to the best of the researchers’ knowledge, only two previous studies discussed the consequences of food insecurity on university students, whereby one study was qualitative and the other focused on anthropometric measurement. As for this study, the consequences of food insecurity on psychosocial factors and academic performance were our main concerns. 

## 5. Conclusions

In conclusion, the large percentage of food insecurity in this study suggests that it is the main concern currently faced by students. A working father, monthly household income, and financial aid recipient were the main predictors for food insecurity among students in this study. The similar trends from previous studies proved that food insecurity is a significant problem to students, and this study revealed that it significantly hinders their wellbeing and academic achievement. With students being the future of our nation, it is very important for the government and higher education institutions to replan the program and policy priorities for students at greater risk of food insecurity. At an individual level, intervention strategies to improve student’s financial management and food literacy can greatly help the students. Furthermore, the food bank or food pantry program provided by some institutions may help students to combat food insecurity if the program is systematically planned and carried out. On the other hand, future researchers can also conduct studies focusing on identifying the mediating factors on food insecurity towards student’s wellbeing and success so that they can be used in intervention programs, thus improving the students’ futures.

## Figures and Tables

**Table 1 ijerph-18-05627-t001:** Characteristics of respondents.

Variables	*n* (%)	Mean ± SD
Age (years)		21.98 ± 1.122
Gender		
Male	215 (32.4)	
Female	448 (67.6)	
Living arrangement		
On/Off Campus	639 (96.4)	
With family	24 (3.6)	
Marital Status		
Single	638 (96.2)	
Married	25 (3.8)	
Father occupation		
Working	415 (82.5)	
Not working	88 (17.5)	
Mother occupation		
Working	352 (64.1)	
Not working	180 (35.9)	
Monthly household income ^1^ (MYR *)		6746.65 (=USD 1623.52) ± 5487.56
<MYR 4850 (=USD 1164)	208 (44.6)	
MYR 4850 (=USD 1164) -MYR 10,959 (=USD 2630.16)	204 (43.8)	
>MYR 10,959 (=USD 2630.16)	54 (11.6)	
Financial aid recipient		
Yes	592 (91.4)	
No	56 (8.6)	
Estimation of total financial received (MYR *)		2801.39 (=USD 672.33) ± 1393.44
Estimation of total financial expenses (MYR *)		1718.78 (=USD 412.50) ± 876.46
Food security status		
Food-secure	244 (37.2)	
Food-insecure	412 (62.8)	
CGPA ^2^		3.42 ± 0.208
<3.7	534 (88.6)	
≥3.7	69 (11.4)	
Stress		25.50 ± 15.178
Normal	434 (65.5)	
Moderate	181 (27.3)	
Severe	48 (7.3)	
Anxiety		25.82 ± 14.623
Normal	150 (22.7)	
Moderate	279 (42.3)	
Severe	231 (35.0)	
Depression		17.79 ± 15.494
Normal	396 (59.9)	
Moderate	220 (33.3)	
Severe	45 (6.8)	

^1^ Income based on thresholds of monthly household gross income Malaysia 2019. * MYR 1 (=USD 0.24). ^2^ Cumulative grade point average (CGPA).

**Table 2 ijerph-18-05627-t002:** Associations between demographic and socioeconomic, psychosocial factors and academic performance, and food security status.

Variables	Food Secure	Food Insecure	χ^2^	*p*
Gender			0.002	0.962
Male	80 (32.7)	133 (32.3)		
Female	164 (67.3)	279 (67.7)		
Living arrangement			2.215	0.137
On/Off Campus	232 (95.1)	402 (97.6)		
With family	12 (4.9)	10.(2.4)		
Status			0.621	0.434
Single	237 (97.1)	389 (95.6)		
Married	7 (2.9)	18 (4.4)		
Father occupation			3.866	0.049 *
Working	153 (87.4)	26 (28.6)		
Not working	22 (12.6)	65 (71.4)		
Mother occupation			5.086	0.024 *
Working	124 (70.9)	194 (60.2)		
Not working	51 (29.1)	128 (39.8)		
Monthly household income			9.234	0.010 *
<MYR 4850 (=USD 1164)	26 (31.3)	78 (53.8)		
MYR 4850 (=USD 1164) -MYR 10,959 (=USD 2630.16)	45 (54.2)	53 (36.6)		
>MYR 10,959 (=USD 2630.16)	12 (14.5)	14 (9.6)		
Financial aid receiver			12.319	<0.001 **
Yes	210 (86.1)	375 (94.5)		
No	34 (13.9)	22 (5.5)		
Dependent variables:				
CGPA			14.601	<0.001 **
<3.7	198 (82.2)	330 (92.7)		
≥3.7	43 (17.8)	26 (7.3)		
Stress			8.733	0.015 *
Normal	174 (71.3)	253 (61.4)		
Moderate	59 (24.2)	121 (29.4)		
Severe	11 (4.5)	38 (9.2)		
Anxiety			38.936	<0.001 **
Normal	86 (35.2)	62 (15.2)		
Moderate	99 (40.6)	184 (45.0)		
Severe	59 (24.2)	163 (39.8)		
Depression			39.030	<0.001 **
Normal	182 (74.6)	205 (50.0)		
Moderate	49 (20.1)	173 (42.2)		
Severe	13 (5.3)	32 (7.8)		

* Significant at *p* < 0.05. ** Significant at *p* < 0.001.

**Table 3 ijerph-18-05627-t003:** Factors associated with food security status and its consequences towards psychosocial factors and academic performance.

Variables	B	Adjusted OR (95% CI)	*p*
Factors associated with food insecurity: Father occupation			
Working	1.864	6.446 (1.22–34.014)	0.028 *
Not working		Ref	
Mother occupation			
Working	0.697	2.008 (0.650–6.203)	0.226
Not working		Ref	
Monthly household income			
<MYR 4850 (=USD 1164)	2.661	14.314 (1.565–130.954)	0.018 *
MYR 4850 (=USD 1164) -MYR 10,959 (=USD 2630.16)	2.753	15.687 (1.720–143.092)	0.015 *
>MYR 10,959 (=USD 2630.16)		Ref	
Financial aid receiver			
Yes	1.034	2.811 (1.602–4.932)	<0.001 **
No		Ref	

* Significant at *p* < 0.05. ** Significant at *p* < 0.001. Note: B = standardized regression coefficients, OR = odds ratio, CI = confidence interval.

**Table 4 ijerph-18-05627-t004:** Food security status affect psychosocial status and academic performance.

Variables	B	Adjusted OR (95% CI)	*p*
Consequences of food insecurity: CGPA			
≥3.7	−1.014	0.363 (0.216–0.609)	<0.001 **
<3.7		Ref	
Stress			
Moderate and Severe	0.446	1.562 (1.111–2.192)	0.010 *
Normal		Ref	
Anxiety			
Moderate and Severe	1.114	3.046 (2.090–4.441)	<0.001 **
Normal		Ref	
Depression			
Moderate and Severe	1.077	2.935 (2.074–4.151)	<0.001
Normal		Ref	

* Significant at *p* < 0.05. ** Significant at *p* < 0.001. Note: B = standardized regression coefficients, OR = odds ratio, CI = confidence interval.

## Data Availability

The data presented in this study are available on request from the corresponding author. The data are not publicly available due to privacy and ethical restrictions.

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
