# Peer review of "Food Insecurity: Is It a Threat to University Students’ Well-Being and Success?"

_ijerph, 2021, doi:10.3390/ijerph18115627_

Round 1
Reviewer 1 Report
Referee Report: “Food insecurity: it is a threat to university students’ wellbeing and success”
(ijerph-1188283)
The study aims to determine the factors associated with food insecurity among Malaysian university students as well as whether food insecurity is a predictor of students’ academic performance and psychology measured by stress, anxiety, and depression. Using a sample of 663 undergraduate students, the authors find almost 63% of students are food insecure. They report that students with working fathers, and from low and middle income families, are more likely to be food insecure. On the other hand, food insecure students are less likely to have a cumulative GPA exceeding 3.7 while they are more likely to be stressed, anxious, and depressed. One of the principal contributions of this paper is the focus on University students and their food security status in Malaysia.
The paper uses primary data from an online survey of students selected randomly from different faculties to contribute to the growing literature on food insecurity on University campuses across the world. However, the paper needs to be thoroughly edited. My comments are below.
Comments:
- Overall, I think it’s important for the authors to address all typos and grammatical errors in the draft. These errors often hinder easy reading. I recommend the authors use an editor. For example, the title of the paper: Food insecurity: it is a threat to university student’s wellbeing and success?” should be edited to say either “Food insecurity: it is a threat to university students’ wellbeing and success” or Food insecurity: is it a threat to university students’ wellbeing and success?” There are many such minor errors that need to be addressed throughout the draft.
- On line 45: It would be good to state the national prevalence rate of food insecurity in Malaysia.
- Please clarify the sentence on lines 63-64.
- In general, we state consequences of food insecurity on students’ health and success and not towards students’ health and success. (Line 66)
- Lines 67-71: The principal contribution of the paper needs to be clarified here. There are 2 research questions that the authors are trying to answer: 1) Student characteristics associated with food insecurity, and 2) Is food insecurity a predictor of CGPA and psychological wellbeing? The first research question is addressed in Section 3.1 and Table 3. The second research question is addressed in Section 3.2 and Table 4.
- When discussing the results, it would be good to reiterate these research questions so that readers can link the results to the research questions posed in the Introduction.
- Table 2 needs to be relabeled as “Associations between demographic and socioeconomic, psychosocial factors and academic performance, and food security status.”
- In lines 103-104: Please elaborate on the cutoff definitions of normal, moderate, and severe.
- In lines 105-106: Please mention the CGPA was converted to a binary measure for the analysis and explain the cut-off choice of 3.7. The cut-off maybe obvious to readers who are familiar with the Malaysian academic system, but for an unfamiliar reader, it would be good to explain it.
- What is B in Table 3 and Table 4?
- In Line 205: please replace financial aid receiver with financial aid recipient.
- In Line 206: Please replace abroad with foreign.
Reviewer 2 Report
Thank you for the opportunity to read this manuscript. The topic is of interest and important to be discussed among government and universities.
I have some important comments to improve the manuscript.
First, the title brings a question so it needs to be: Is it a treat...
Also it bring well-being in the title, but well-being is not brought inside the introduction and also discussion.
Introduction
Lin 55 - "the growing literature ... Where are are the citations?
define well-being and its relation to stress, anxiety and depression
Methods
Line 87 - change the sentence - informed consent "formed"
Is UPM a public or private university?
How many courses are there? Why did you randomly selected 7? Does these 7 courses represent the university?
Why did you choose only students from 2nd and 3rd years?
Which are these 7 courses? are any of them from the health area? It is important especially if one of the courses is related to nutrition.
What is the type of financial aid received by the students? monthly? how much? is there a criteria for receiving it?
Does the university have a university restaurant? does it have different prices according to the student´s income?
In this section authors say they asked about the total financial amount of the aid. nothing is discussed or presented in the following sections.
Was the academic performance checked with the university system?
Results
Line 118 - take out were from the sentence
was the monthly household income related to their father´s and mother´s, right? I am asking because more than 90% live outside their houses, so I wonder how this question was conducted to make sure they were answering about their parents' income.
Why was the CGPA cut at 3.7?
line 139 - rewrite associated
Discussion
As cited in reference 20, did researchers ask the students the type of works their parents perfomed?
What are the criterion to receive financial aid at UPM?
line 233 - take out "that" from the sentence
authors use well-being, is it only related to stress, anxiety and depression?
discuss the concept of well-being
More discussion is needed for FI and financial aid. what are the programs at UPM to help students?
Conclusions are fine.
Author Response
Dear reviewer,
Please see the attachment below. Thank you

Round 2
Reviewer 2 Report
Authors have improved the manuscript answering my previous questions. However part of the answers were not include inside the manuscript.
Point 7: “Methods: How many courses are there? Why did you randomly selected 7? Does these 7 courses represent the university?”
Part of this answer should be included in the manuscript so readers can understand the selection.
Point 8: “Methods: Why did you choose only students from 2nd and 3rd years?”
a summary of this answer should also be included
Point 9: “Methods: Which are these 7 courses? are any of them from the health area? It is important especially if one of the courses is related to nutrition.”
The name of the courses should also be included.
“Methods: What is the type of financial aid received by the students? monthly? how much? is there a criteria for receiving it?”
This question was not fully answered.
Besides these last comments, the manuscript had improved.
Author Response
Dear reviewer,
Please see the attachment below. Thank you.
